# Creating amphiphilic porosity in two-dimensional covalent organic frameworks via steric-hindrance-mediated precision hydrophilic-hydrophobic microphase separation

Shu-Yan Jiang[1,2,3], Zhi-Bei Zhou[1,2,3], Shi-Xian Gan[1], Ya Lu[1], Chao Liu[1], Qiao-Yan Qi[1], Jin Yao [1] ✉ & Xin Zhao [1,2] ✉

Creating different pore environments within a covalent organic framework (COF) will lead to useful multicompartment structure and multiple functions, which however has been scarcely achieved. Herein we report designed synthesis of three two-dimensional COFs with amphiphilic porosity by steric-hindrance-mediated precision hydrophilic-hydrophobic microphase separation. Dictated by the different steric effect of the substituents introduced to a monomer, dual-pore COFs with kgm net, in which all hydroxyls locate in trigonal micropores while hydrophobic sidechains exclusively distribute in hexagonal mesopores, have been constructed to form completely separated hydrophilic and hydrophobic nanochannels. The unique amphiphilic channels in the COFs enable the formation of Janus membranes via interface growth. This work has realized the creation of two types of channels with opposite properties in one COF, demonstrating the feasibility of introducing different properties/functions into different pores of heteropore COFs, which can be a useful strategy to develop multifunctional materials.

Covalent organic frameworks are a class of crystalline organic porous polymers which are constructed by assembling organic building blocks through covalent bonds[1,2]. They have been exploited as versatile functional materials for many fields including gas storage[3–6], membrane separation[7–10], catalysis[11–14], sensing[15–19], capture of hazardous materials[20–24], and energy storage[25–28]. As a class of porous materials, the pore structure of a COF is a key foundation on which its application is explored. In this context, pore engineering has been developed to fabricate COFs with different pore structures, or to tune pore environment via post-synthetic modification, through which their functions and applications are established[29–33]. In most cases, the pore structure in a COF is isomorphic because most COFs reported so far possess homogeneous porosity, that is, a COF holding only one type of pores. As a result, their pore property/function is also homogeneous.

Integrating different kinds of pores into one COF will lead to a well-arranged multicompartment architecture due to its extended crystalline structure. Such structure can offer distinct features such as exhibiting different properties in each isolated part, having the capability of separate storage of different types of molecules (even incompatible ones), and implementing different functions in different

[1]Key Laboratory of Synthetic and Self-Assembly Chemistry for Organic Functional Molecules, Center for Excellence in Molecular Synthesis, Shanghai Institute of Organic Chemistry, Chinese Academy of Sciences, 345 Lingling Road, 200032 Shanghai, China. [2]University of Chinese Academy of Sciences, 100049 Beijing, China. [3]These authors contributed equally: Shu-Yan Jiang, Zhi-Bei Zhou. ✉e-mail: yaojin@ucas.ac.cn; xzhao@sioc.ac.cn

kinds of pores, which can give rise to multifunctional materials. Multicompartment plays a crucial role in biological systems and exhibits potential applications in biomedicine[34,35], while multifunctional materials are an important basis for the fabrication of sophisticated systems, for instances, cooperative catalysis[36,37], co-delivery[38–40], and selective separation[41]. However, although COFs with heterogeneous porosity (termed heteropore COFs) have emerged and drawn increasing attention over the past several years[42,43], their studies are mainly focused on developing novel structures[44–49]. The unique property or function endowed by heteroporous structures has been rarely demonstrated[36,50]. Especially, introducing different properties into different kinds of pores of a heteropore COF is hardly achieved due to the challenge of controlling the orientation of functional units in specified pores.

Herein, we propose a strategy to construct amphiphilic COFs with separated hydrophilic and hydrophobic nanochannels by precisely controlling the distribution of hydrophilic and hydrophobic segments in two different kinds of pores, respectively. The design is based on a dual-pore COF with kgm lattice which consists of alternate and periodical arrangement of trigonal and hexagonal pores in a plane[51]. As demonstrated by our previous work, the small trigonal micropores can accommodate three hydroxyl groups but not substituents of larger size due to the steric hindrance[52]. Based on this result, we proposed to regulate the orientation of two substituents by their difference in steric effect. As shown in Fig. 1a, when an asymmetrically substituted dialdehyde is polymerized with tetraethylenetetramine (ETTA) to produce a dual-pore COF with kgm lattice, the large OR group (R is butyl, benzyl or naphthylmethyl) cannot be accommodated in trigonal micropores due to steric hindrance and thus can only locate in hexagonal mesopores, which forces the smaller hydroxyl group to exclusively distribute in trigonal pores. It will lead to the formation of 2D COFs with amphiphilic porosity[53], enabling precision microphase separation of hydrophilic region (trigonal channels) and hydrophobic region (hexagonal channels) in the COFs.

## Results

### Synthesis and characterization of COF-R

COF-Bu was firstly synthesized through the polycondensation of ETTA and 2-butoxy-5-hydroxyterephthalaldehyde (TPA-Bu) under a solvothermal condition (see detail in Section B in the Supplementary Information). The Fourier transform infrared (FT-IR) spectrum of the as-obtained crystallites revealed a peak at 1617 cm$^{-1}$ which is assignable to the newly formed C = N bond, while the bands corresponding to C = O (1674 cm$^{-1}$) from TPA-Bu and −NH$_2$ (~3300 cm$^{-1}$) from ETTA almost vanished, indicating completion of the condensation reaction (Supplementary Fig. 1). The formation of polyimine structure is also supported by the appearance of resonance signal of C = N units at 159.9 ppm in its solid-state $^{13}$C CP/MAS NMR spectrum (Supplementary Fig. 2). Furthermore, four signals corresponding to *n*-butoxyl chains are observed in the region of 10–70 ppm. These results essentially verify the formation of imine-linked COF in terms of chemical composition and connection. Thermogravimetric analysis (TGA) shows that COF-Bu has excellent thermal stability, as supported by only 5% weight loss when temperature increases to 409 °C (Supplementary Fig. 3). Scanning electron microscopy (SEM) images show COF-Bu consists of irregular particles with shale-like morphology (Supplementary Fig. 4a, b). Under transmission electron microscopy (TEM), most area in COF-Bu was clearly observed to have an abundant ordered lattice texture and these particles display sheet-stacked morphology, suggesting the formation of a 2D structure with high crystallinity (Supplementary Fig. 4c, d).

The crystal structure of COF-Bu was resolved by powder X-ray diffraction (PXRD) analysis and theoretical simulations. COF-Bu exhibited high crystallinity with a series of diffraction peaks at 2.63°,

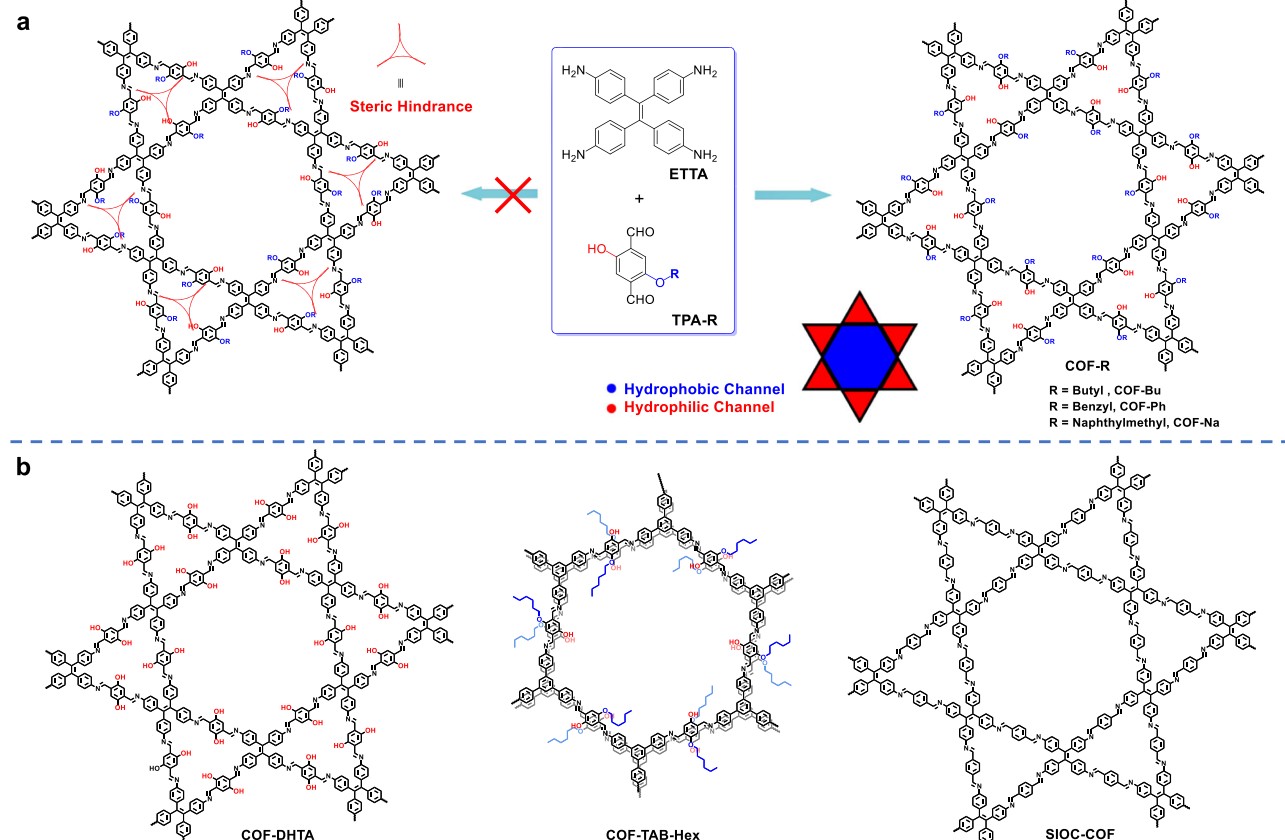

**Fig. 1 | Schematic diagram. a** Strategy for steric-hindrance-mediated microphase separation in COF-R. **b** Structures of the COFs used for comparison studies.

5.27°, 6.98°, and 7.92°, which are assigned to (100), (200), (210) and (300) facets, respectively. Another two weak peaks assignable to (310) and (400) facets are also observed at 9.50° and 10.56°, respectively (Fig. 2a). Lattice modeling and Pawley refinement of COF-Bu were conducted with Materials Studio version 7.0. The experimental data matched well with the calculated PXRD pattern for a dual-pore COF (kgm net) with eclipse interlayer stacking (Fig. 2). Pawley refinement afforded unit-cell parameters of $a = b = 38.75$ Å, $c = 5.31$ Å, $\alpha = \beta = 90°$ and $\gamma = 120°$, with $R_{wp} = 3.77\%$ and $R_p = 1.94\%$ (Supplementary Table 1). Other possible 2D structures including dual-pore framework with staggered stacking and single-pore isomer were ruled out by the mismatches between the simulated and experimental PXRD patterns (Fig. 2d and Supplementary Fig. 5).

The hierarchical porosity of COF-Bu was further verified by the result from nitrogen adsorption-desorption measurement. The $N_2$ isotherm shows a stepwise adsorption, with a sharp uptake before $P/P_0 = 0.01$ and a slower adsorption between $P/P_0 = 0.05–0.15$ (Fig. 3a), indicating that COF-Bu possesses both microporous and mesoporous characters. Its Brunauer-Emmett-Teller (BET) surface area was calculated to be 1031 $m^2/g$ (Supplementary Fig. 6a) and a total pore volume (at $P/P_0 = 0.99$) of 0.77 $cm^3/g$ was obtained. Pore size distribution (PSD) analysis reveals two narrow distributions at 6.3 and 24.5 Å, respectively (Fig. 3b). Compared with its homologue framework COF-DHTA (Fig. 1b), the smaller pore of COF-Bu is perfectly matched with the trigonal pore of COF-DHTA in size (6.3 Å)[52], while the bigger pore is

smaller than that of COF-DHTA (26.5 Å) due to the presence of the *n*-butoxyl chains. Meantime, the experimentally observed pore size of the small pore well matches its theoretical value (6.3 Å), while there is a significant difference between the experimental (24.5 Å) and the theoretical (18.6 Å) pore size of the large pore (Supplementary Fig. 6b). This could be attributed to the flexibility of the butyl groups. These results strongly suggest that hydroxyl groups are in the trigonal micropores and *n*-butoxyl chains distribute in the hexagonal mesopores, further indicating the successful construction of the expected heteropore COF with amphiphilic porosity.

To provide more comprehensive evidence to the feasibility of this strategy, we chose two larger R groups (benzyl and naphthylmethyl groups), considering that the size of the butyl group might not completely rule out the possibility of its partial distribution in the small triangle pore. The increase in the size of R groups further enlarged the steric hindrance difference in the distribution of groups within the two types of pores. When the R group is naphthylmethyl, the possibility of it entering the small pore is completely ruled out, as its size is too large to be accommodated in the triangular pore. As a proof of concept, COF-Ph and COF-Na were synthesized (see Section B in the Supplementary Information) and subsequently characterized using FT-IR spectroscopy and solid-state $^{13}C$ CP/MAS NMR spectroscopy. The bands corresponding to C = N at 1614 $cm^{-1}$ were observed for both COF-Ph and COF-Na in their FT-IR spectra and the signals belonging to the carbon of C = N appeared at 155.3 and 152.7 ppm in the $^{13}C$ NMR

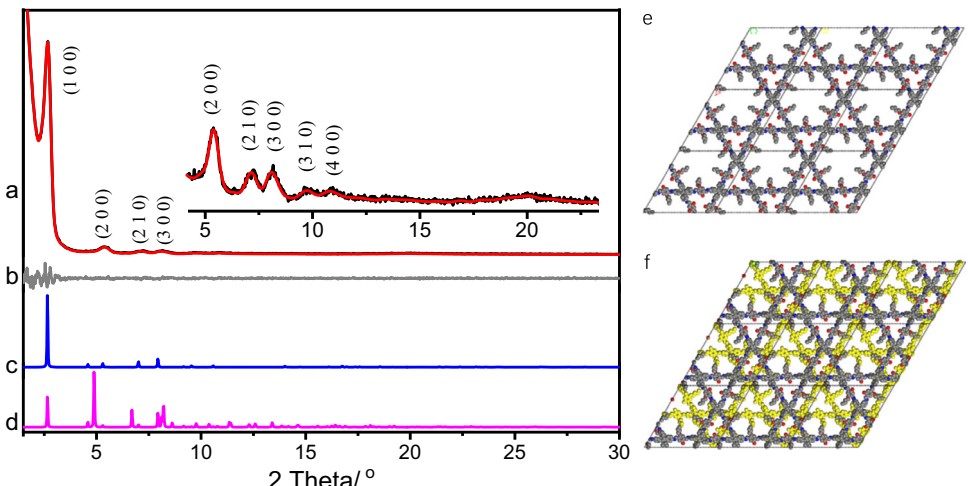

**Fig. 2 | Structural characterization. a** Experimental (black) and refined (red) PXRD patterns of COF-Bu. **b** Difference plot between the experimental and refined PXRD patterns. Simulated PXRD patterns for (**c**) eclipsed and (**d**) staggered dual-

pore structures. Structural representation of COF-Bu with (**e**) eclipsed and (**f**) staggered stacking.

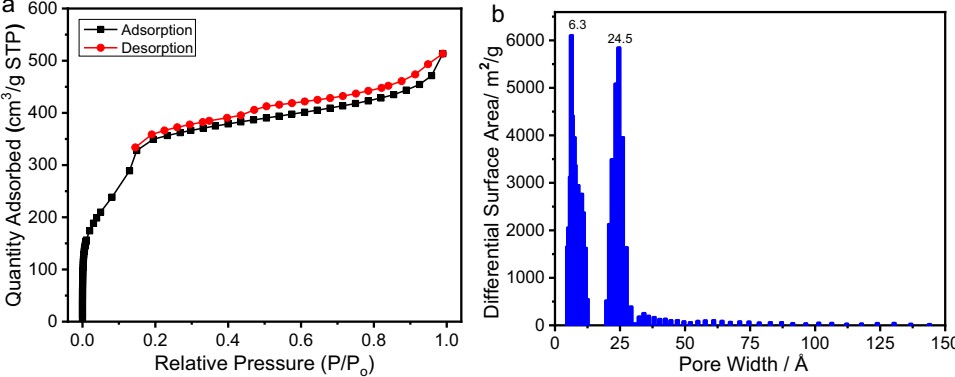

**Fig. 3 | Porosity characterization. a** $N_2$ adsorption-desorption isotherm at 77 K. **b** pore size distribution profile of COF-Bu.

spectra for COF-Ph and COF-Na, respectively (Supplementary Figs. 7, 8 and Figs. 13, 14). These results verified the successful formation of polyimine structure for COF-Ph and COF-Na. TGA shows that COF-Ph and COF-Na also have excellent thermal stability, as only 5% weight loss happened for the two COFs when the temperature reached 345 and 380 °C, respectively (Supplementary Figs. 9, 15). The microscopic morphology of COF-Ph and COF-Na was also investigated by SEM (Supplementary Figs. 10, 16). COF-Ph powder exhibits a morphology of smooth, rhombic plate shaped, and flower-like nanostructures, while COF-Na powder exhibits irregular damaged vesicle-like structure.

Structural simulations indicated that the experimental PXRD patterns of COF-Ph and COF-Na agreed with the simulated PXRD patterns of the dual-pore structure (Supplementary Figs. 11, 17). For COF-Ph, the main diffraction peak was observed at 2.67°, corresponding to the (100) facet, while the same facet peak showed at 2.74° for COF-Na. Other weak peaks at 5.39° and 7.10° were assigned to (200) and (210) facets for COF-Ph, while peak at 5.37° was assigned to (200) facet for COF-Na. All the experiment results fit well with the simulated patterns with eclipsed interlayer stacking structures, indicating successful construction of the expected dual-pore COFs. Pawley refinement afforded unit-cell parameters of $a = b = 38.68$ Å, $c = 5.67$ Å, $\alpha = \beta = 90°$, $\gamma = 120°$; $R_{wp} = 5.90\%$, $R_p = 3.52\%$ for COF-Ph, and $a = b = 38.66$ Å, $c = 5.40$ Å, $\alpha = \beta = 90°$, $\gamma = 120°$; $R_{wp} = 7.31\%$, $R_p = 4.69\%$ for COF-Na (Supplementary Tables 2 and 3).

$N_2$ adsorption measurements were carried out at 77 K to access porosity parameters of COF-Ph and COF-Na. Their $N_2$ isotherms are similar to that of COF-Bu, which both have stepwise adsorption behavior. Their Brunauer-Emmett-Teller (BET) surface areas were calculated to be 720 m²/g for COF-Ph and 476 m²/g for COF-Na, (Supplementary Figs. 12, 18). Their total pore volumes were calculated to be 0.65 cm³/g (COF-Ph) and 0.35 cm³/g (COF-Na) at $P/P_0 = 0.99$. PSD analysis indicates that these two COFs exhibit similar dual-pore distributions (6.0 and 18.8 Å for COF-Ph, 7.3 and 17.1 Å for COF-Na) and their experimental pore sizes match well with their theoretical values (7 and 19 Å for COF-Ph, 7 and 17 Å for COF-Na). A comparison of these data could provide more information for the distribution of the substituents in the three COFs. Taking COF-Na as a reference, since naphthylmethyl is too large to enter the triangular pores, they can only locate in the hexagonal mesopores, which results in exclusive distribution of all the hydroxyl groups in the triangular pores. The experimentally observed size of the triangular pore in the three COFs are quite close. The difference within 1 Å may be attributed to the increase in the volume of R groups, leading to different rotational dihedral angles of the TPA benzene ring segments within the different frameworks. Their similar pore size suggests that the three COFs have a similar triangular pore environment, again confirming the precise control of the microenvironment in different pores of these hetero-pore COFs via steric hindrance to create amphiphilic porosity.

## Synthesis and characterization of the COF-R membranes
With these amphiphilic heteropore COFs available, an investigation on their distinctive property was conducted. Since COF powder is bulk material and thus macroscopically isotropic, properties resulted from the hydrophilic-hydrophobic microphase separation are difficult to reflect. Therefore, confined growth of COFs at an interface is used. To this end, COF-Bu and COF-Ph membranes were fabricated at an interface of water and an immiscible organic solvent, which afforded freestanding membranes. By comparing PXRD patterns and FT-IR spectra, the structural consistency between the powder and the membranes of COF-Bu and COF-Ph were confirmed (Section D in the Supplementary Information). Contact angle (CA) analysis was used to study the wettability of these membranes. It was found that the CA of the surface contacting water (upper surface) of the COF-Bu membrane was 82° and the CA of the surface contacting organic solvent (bottom surface) was 130° (Fig. 4a, b). A similar phenomenon was also observed for the membrane of COF-Ph, with CAs of its upper and bottom surfaces being 47° and 99°, respectively (Supplementary Fig. 39). These results indicate that under the induction of the interface, a hydrophilic-hydrophobic difference is generated between the upper and bottom surfaces of the membranes, affording generalized Janus membranes.

## Synthesis and characterization of other COFs and membranes for comparison
To investigate if it is the separated hydrophilic and hydrophobic nanochannels in COF-Bu and COF-Ph that leads to the formation of the Janus membranes, another two COFs were synthesized for comparison (Fig. 1b): one is a heteropore COF without hydrophilic and hydrophobic substituents on the linker (SIOC-COF), and the other is a single-pore COF (COF-TAB-Hex) constructed from a linker similar to that of COF-Bu (The detailed characterizations of these COFs are shown in Section E and Supplementary Tables 4, 5 in the Supplementary Information). It should be noted that although COF-TAB-Hex bears both hydroxyl groups and n-hexoxyl chains on its linker, it does not hold amphiphilic porosity because of random distributions of the two substituents in its channels. Membranes corresponding to the two COFs were also prepared under a similar interfacial polymerization condition as that for the COF-Bu membrane (Supplementary Figs. 33–35). In addition, a POP-Bu membrane, which has the same chemical composition as the COF-Bu membrane but without crystallinity, was also prepared (Supplementary Figs. 36, 37). Its amorphous

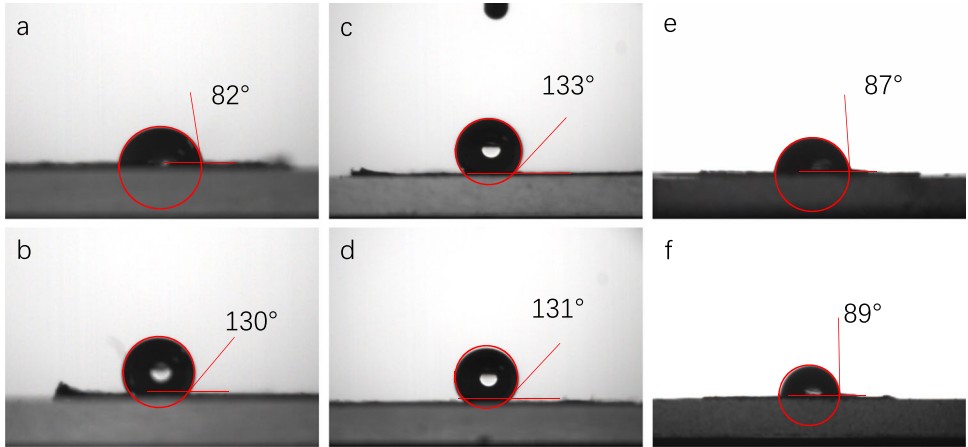

**Fig. 4 | Contact angles. (a)** upper surface and **(b)** bottom surface of the COF-Bu membrane; **(c)** upper surface and **(d)** bottom surface of the COF-TAB-Hex membrane; **(e)** upper surface and **(f)** bottom surface of the SIOC-COF membrane.

structure makes it lack of amphiphilic porosity. For the COF-TAB-Hex membrane (Fig. 4c, d), the contact angles of the upper and bottom surfaces are almost the same (133° and 131°). For the SIOC-COF membrane (Fig. 4e, f), the contact angles of both surfaces are also essentially the same (87° and 89°). In the case of the POP-Bu membrane, the contact angles of the upper and bottom surfaces are also very close (86° and 87°, Supplementary Fig. 42). The results suggest that the hydrophilic-hydrophobic microphase separation in COF-Bu and COF-Ph is the key to endow the membranes with the Janus property.

### Proposed mechanism for the formation of Janus COF Membranes

The morphology of the membranes was investigated with SEM. For COF-Bu, the upper surface displays a smooth planar structure, but its bottom surface is quite rough by distribution of sphere-like particles (Supplementary Fig. 38). For COF-Ph, the upper surface displays a surface with snowflake-like bumps and scattered small sphere-like particles on a relatively smooth surface, while the bottom surface displays a surface with many irregular aggregates (Supplementary Fig. 39). Such morphology was not observed for the SIOC-COF membrane, which displays similar rough planar morphology for both the upper and bottom surfaces (Supplementary Fig. 40). In the case of the COF-TAB-Hex membrane, SEM revealed that both the upper and bottom surfaces consist of aggregation of rod-like structures and irregular particles (Supplementary Fig. 41). In the case of POP-Bu membranes, the upper and bottom surfaces clearly display some difference, exhibiting different roughness (Supplementary Fig. 42). These results indicates that difference in surface morphology of the five membranes does not play a crucial role in determining their wettability, again suggesting that it is the amphiphilic porosity of COF-Bu and COF-Ph that leads to different surface property during the interfacial polymerization. In this aspect, the difference was proposed to be probably related to the following reasons: basically, the orientation of the hydroxyl and the OR groups could be induced by the interface between the two solvents with opposite polarity, for which the hydrophilic hydroxyl groups tend slightly point to the water side while the hydrophobic OR chains face to the organic solvent side. For example, in COF-Bu, the hydroxyl groups and butoxyl chains are arranged in highly ordered arrays in their respective channels. Therefore, the direction of the substituents in the adjacent layers is synclastic and ordered (Fig. 5a). As a result, the wettability of the two surfaces becomes different. For COF-TAB-Hex, it has only one type of pores in which the hydroxyl groups and $n$-hexoxyl chains are randomly distributed. Although the hydroxyl groups and alkyl chains could also experience similar induction effects, their random distributions will unavoidably result in opposite orientations of the substituents in the adjacent layers (Fig. 5b), which results in steric hindrance between them and thus weakens the induction. Consequently, both the two surfaces exhibit the similar wettability. For the SIOC-COF membrane, it does not have amphiphilic building blocks and thus induction by interface does not work. In the case of the POP-Bu membrane, due to its amorphous structure, control over the distribution of side chains is no longer possible, which leads to loss of amphiphilic porosity and synclastic induction effect.

## Discussion

In summary, through sterically controlling the orientation of substituents with different sizes and polarities in the pores, three heteropore 2D COFs with amphiphilic porosity have been constructed. The dual-pore COFs, holding kgm net, are assembled by arranging small hydroxyl groups in their trigonal micropores and large substituents in their hexagonal mesopores. As a result, well-ordered hydrophilic microchannels and hydrophobic mesochannels are built separately within the COFs, which leads to a precision hydrophilic-hydrophobic microphase separation. A distinctive feature endowed by this unique amphiphilic porosity has been further demonstrated by the formation of self-supporting COF membranes with Janus property from interfacial polymerization. The successful construction of COFs with opposite channel environments suggests that COFs with heteroporosity have a great potential for developing multiple-property/function materials, due to the structural advantage that different kinds of pores can be modified independently, from which functions different from or superior to those based on COFs with homogeneous porosity may be expected.

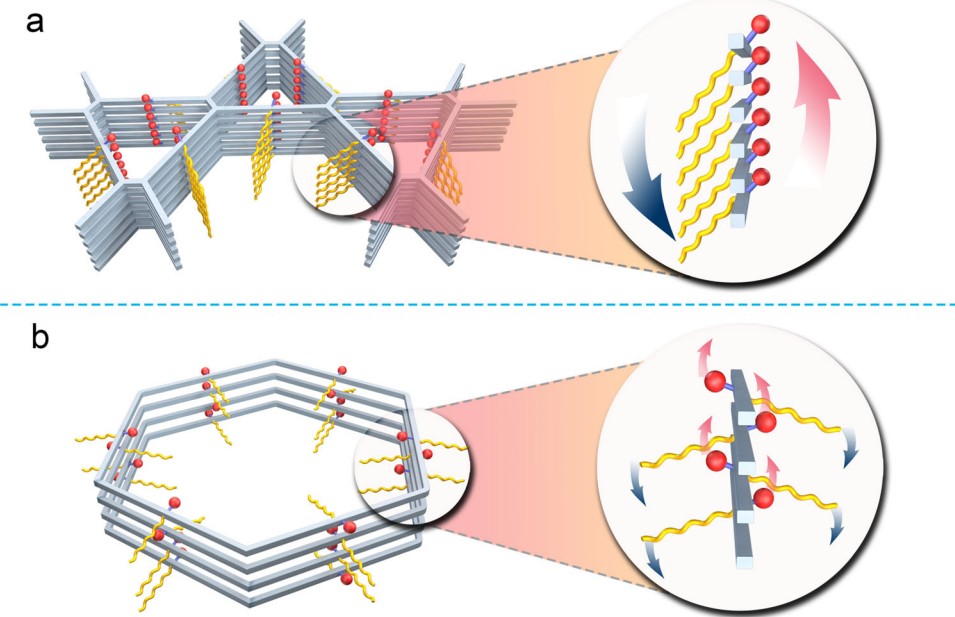

**Fig. 5 | Proposed mechanism.** The explanations for the different surface wettability of the membranes fabricated from (**a**) COF-Bu and (**b**) COF-TAB-Hex at water-dichloromethane interfaces.

## Methods

### Synthesis of the COFs

Generally, all the COFs were synthesized by the solvothermal method in Pyrex tubes at 120 °C for 3 days. A glass ampoule (10 mL) was charged with the monomers and specific solvents. The tube was first sonicated for 10 min and then an aqueous acetic acid solution (3 or 6 M) was added as a catalyst. The ampoule was sealed after being degassed in a liquid nitrogen bath for 5 min, warmed to room temperature and then kept at 120 °C without disturbance for 3 days. The as-formed precipitate was filtered and washed with THF and acetone several times. Finally, the COF powders were dried under dynamic vacuum. The specific combinations of the solvents are given in Section B in the Supplementary Information.

### General procedure for the preparation of the membranes

The membranes are prepared by the interfacial polymerization method. First, the aldehyde monomer (102 µmol) was added in dichloromethane (20 mL) in a glass beaker (50 mL) and treated with ultrasound to afford a clear solution (aldehyde solution). Then, a solution of amine monomer (51 µmol) and AcOH (1 mL) was prepared by adding them to water (19 mL) in a glass beaker (50 mL) and treated with ultrasound (amine solution). The as-prepared amine solution was slowly dripped onto the surface of the aldehyde solution with a disposable syringe to afford a two-phase interface. The system was kept at room temperature for 72 h under an undisturbed condition. It could be observed that the thin layers formed at the interface of water and dichloromethane. The as-prepared COF membranes were collected by an ethanol-wetted Nylon membrane (pore size: 0.1 µm) after removing the top aqueous phase with a dropper and washed with dichloromethane, tetrahydrofuran, and acetone, respectively. Finally, the residual solvent was removed by treating the COF membranes in a vacuum drying oven under 60 °C. It should be noted that the interfacial polymerization to fabricate the COF-Ph membrane was carried out with water/o-dichlorobenzene phases at 40 °C for three days. The POP-Bu membrane was fabricated via polymerization of ETTA and TPA-Bu with toluene/water phases at room temperature for three days.

## Data availability

All data supporting the findings of this study are available within the article, as well as the Supplementary Information file, or available from the corresponding authors on request.

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

## Acknowledgements
The authors thank the National Science Fund for Distinguished Young Scholars of China (No. 21725404, X.Z.) and the Science and Technology Committee of Shanghai Municipality (20JC1415400, X.Z.) for financial support.

## Author contributions
X.Z. and S.J. conceived the project. X.Z. supervised the project. S.J., Z.Z. and S.G. designed and performed the experiments for the synthesis of the monomers, COFs, and the characterization including FT-IR, ss 13C NMR, PXRD, N2 sorption, TGA, SEM and TEM. C.L. performed the experiments for the synthesis of ETTA. J.Y. and S.J. designed the experiment of the COF membranes, Y.L., S.J. and Z.Z. perform the synthesis, characterization and analysis of PXRD, FT-IR, contact angles and SEM of the COF membranes. Q.Q. aided in the structural simulations. X.Z. and S.J. wrote the manuscript. X.Z., S.J. and Z.Z. revised the manuscript. All the authors discussed the results and commented on the manuscript.

## Competing interests
The authors declare no competing interests.
