## [Peer Review File · Nature Communications]

Creating Amphiphilic Porosity in Two-Dimensional Covalent Organic Frameworks via Steric-Hindrance-Mediated Precision Hydrophilic-Hydrophobic Microphase SeparationReviewer #1 (Remarks to the Author):

please see document attached

In this paper titled “Creating Amphiphilic Porosity in a Two-Dimensional Covalent Organic Framework via Steric-Hindrance-Mediated Precision Hydrophilic-Hydrophobic Microphase Separation”, Zhao and coworkers described a dual-pore COF with distinct pore properties, namely, hydrophilic pore and hydrophobic pore. The team also made a Janus membrane out of this COF. The same team had contributed a lot work on design and synthesis of hetero-pore COFs in their past works. It is exciting to see finally the hetero-properties are introduced into the hetero-pores. However, several critical issue need to be addressed before this work could be published.

1) Formation of the hetero-pore as proposed by the authors is the key achievement of the current work. This reviewer is not convinced yet on the structure with the current data, mainly sorption and solid state CNMR. According to their previous work, three OR groups all sit in the smaller channel is out of consideration. But, this does not rule out the cases of OH/OR = 2/1 or even OH/OR=1/2 combination in the small pore. Will the simulated pore size be a big difference that can be unambiguously ascertained from the experimental sorption results, considering the relative wide pore size distribution shown in Figure 3b? The argument of steric repulsion is reasonable, but the schematic representation in Figure 1a could be some misleading. The OR chain is flexible and the 2D COF stacking with $\sim 3.5 \text{ \AA}$ interspacing will might somehow accommodate 1 or 2 in the smaller pore. Maybe some simulation work could clarify this. Again, the solid state CNMR is not a strong evidence based on the authors’ claiming that “four signals corresponding to n-butoxyl chains are observed”. The chemical shift in the polycrystalline sample is not as sensitive as one observing in solution NMR, often broadened peak are observed with limited resolution. For example, the CNMR spectrum of COF-TAB-Hex also only has 6 set of peaks (Figure S11) for the OR chain but they might just randomly distributed in different pores (that says not necessarily each pore has the same number of OR chains). Is it possible to use 2D NMR to support the authors’ claim? One good example could be found for a MOF paper (Nature 2022 volume 606, pages 706–712).

2) The study on the Janus membrane part seems not complete to me. The authors’ explanation on the wettability of the film’s two sides are too arbitrary. I don’t see how the pore’s hydrophilicity/hydrophobicity transferred to the overall film’s property. According to their proposed mechanism, COF-TAB-Hex should display

similar properties as COF-Bu (the OH and OR groups can still aligned even though being in the same pore). Besides, many other factors influence the surface wettability, such as the surface morphology. Of all the three COF films this paper present, only COF-Bu shows distinct surface morphology on two sides, which also coincides being the only one with Janus property . The roughness of the bottom surface might contribute more into the hydrophobicity than the surface OR chains do (Colloid and Interface Science Communications 46 (2022) 100556). More experiment evidence is necessary to clarify the Janus properties. It is also interesting to see some applications with COF-Bu membrane to show how the hetero-pore functions exert on the performance.

There are some minor issued need to be addressed by the authors.

1) In Supporting Information, Figure 2, the NMR spectra looks missing one piece indicated in the red box (it would be better for the authors to label the figures and tables in SI as Figure Sx and Tale Sx to avoid confusions).

Figure 2. Solid-state ^{13}C CP/MAS NMR spectrum of COF-Bu.

2) The paxrd patters of SIOC-COF powder and membrane look different. There seems be a shoulder on the main diffraction peak for the membrane sample (SI, Figure 20). The author should comment on this.

3) A zoom in C=N vibration area in the FT-IR spectra can better convey the structure information.

Reviewer #2 (Remarks to the Author):

This is an exciting manuscript on "Creating Amphiphilic Porosity in a Two-Dimensional Covalent Organic Framework via Steric-Hindrance-Mediated Precision Hydrophilic-Hydrophobic Microphase Separation." This manuscript has been written well, and the logic behind the work is sound. Therefore, I recommend acceptance per minor revision.

1. The structures depicted in figure 1 do not showcase the possibility of the disorder. The long-chain aliphatic hydrocarbons will not organize symmetrically. Therefore, figure 1 should be modified, and the considerations of disorder should be presented in the figure.

2. Author wrote "r COFs and membranes for comparative. " I think it should be "comparison".

3. I wondered if he has compared the properties with some COFs like TpAZo or other Tp-based COFs. How would these membranes feature?

4. Very little information regarding the SEM and TEM images of the COF or the membranes has been provided. I would request authors to look into that.

Once these aspects have been taken care of, this manuscript could be accepted.

Reviewer #3 (Remarks to the Author):

The communication of Zhao et al. combines a few concepts in the synthesis of COFs: the use of substituents in the linkers to block certain geometries, the use of hydrophobic and hydrophilic building blocks to create materials with both properties (in this case a Janus membrane).

Although these concepts have been discussed in recent literature (both for MOFs and COFs), the creation of a Janus membrane out of one COF structure is rather new.

The authors over-emphasize the fact that the COFs can have a dual pore size distribution: many papers of such systems (typically kgm but also many others) have appeared since then.

The application of these membranes is missing: how stable are they? are they self-standing (if not, one side of the membrane gets lost of course and the material is no longer a janus membrane). So it would be nice if the author could show a demonstration of its applicability.

Point-by-Point Response to Reviewers' Comments

The authors are very grateful to the reviewers for their valuable comments. These comments are very valuable and helpful for revising our manuscript and improving our research. Here we provide a detailed point-by-point response to their comments, and we have edited the Main Manuscript and the Supplementary Information accordingly. The changes made have been highlighted in yellow in the revised manuscript and Supplementary Information.

Reviewer #1:

In this paper titled “Creating Amphiphilic Porosity in a Two-Dimensional Covalent Organic Framework via Steric-Hindrance-Mediated Precision Hydrophilic-Hydrophobic Microphase Separation”, Zhao and coworkers described a dual-pore COF with distinct pore properties, namely, hydrophilic pore and hydrophobic pore. The team also made a Janus membrane out of this COF. The same team had contributed a lot work on design and synthesis of hetero-pore COFs in their past works. It is exciting to see finally the hetero-properties are introduced into the hetero-pores. However, several critical issue need to be addressed before this work could be published.

Response: We are grateful to the reviewer for taking the time to evaluate our work and greatly appreciate the comments.

Comment 1: Formation of the hetero-pore as proposed by the authors is the key achievement of the current work. This reviewer is not convinced yet on the structure with the current data, mainly sorption and solid state CNMR. According to their previous work, three OR groups all sit in the smaller channel is out of consideration. But, this does not rule out the cases of OH/OR = 2/1 or even OH/OR=1/2 combination in the small pore. Will the simulated pore size be a big difference that can be unambiguously ascertained from the experimental sorption results, considering the relative wide pore size distribution shown in Figure 3b? The argument of steric repulsion is reasonable, but the schematic representation in Figure 1a could be some misleading. The OR chain is flexible and the 2D COF stacking with ~ 3.5 Å interspacing will might somehow accommodate 1 or 2 in the smaller pore. Maybe some simulation work could clarify this. Again, the solid state CNMR is not a strong evidence based on the authors' claiming that “four signals corresponding to n-butoxyl chains are observed”. The

chemical shift in the polycrystalline sample is not as sensitive as one observing in solution NMR, often broadened peak are observed with limited resolution. For example, the CNMR spectrum of COF-TAB-Hex also only has 6 set of peaks (Figure S11) for the OR chain but they might just randomly distributed in different pores (that says not necessarily each pore has the same number of OR chains). Is it possible to use 2D NMR to support the authors' claim? One good example could be found for a MOF paper (Nature 2022 volume 606, pages 706–712).

Response: Thanks for your comments. Based on your concerns, we have conducted additional experiments and analyses. Below please find the details.

(1) The possibility that the smaller pores might accommodate one or two butoxy chains could be reasonably excluded not only by the agreement between the experimental and the simulated pore size distributions of the triangular pores (Figure R1), but also the consistence with the triangular pore of its homologue framework COF-DHTA (Figure R2). There would be a big difference in the experimental pore size if OBU/OH = 1/2 or 2/1 are accommodated within the triangular pores. More specifically, the pore size of the triangular channel in COF-Bu would be much smaller than 6.3 Å if one or two butoxy chains entered.

Figure R1. Schematic diagram of theoretical pore size of COF-Bu.

Figure R2. Structure of COF-DHTA and its pore size distribution profile. (Picture copied from 10.1039/C6SC05673C)

To provide more solid evidence, we have carried out the synthesis of another two dual-pore COFs with larger pendant groups (benzyl and naphthylmethyl), which are named COF-Ph and COF-Na, respectively (Scheme R1). For your reference, we have shown their primary structural characterization data below (Figures R3-6). Taking COF-Na as a reference, since naphthylmethyl is too large to enter the triangular pores, they can only locate in the hexagonal mesopores, which results in exclusive distribution of all the hydroxyl groups in the triangular pores. It is found that the pore size of triangular channels in COF-Ph and COF-Na is consistent with that in COF-Bu, indicating that the three COFs have the same triangular pore environment. These results validate that the smaller hydroxyl groups are all in the triangular micropores, while the larger pendant groups (butyl, benzyl, and naphthylmethyl) exclusively distribute in the hexagonal mesopores.

Scheme R1. The synthesis of COF-Ph and COF-Na.

Figure R3. Experimental and simulated PXRD data and Pawley refinement for (a) COF-Ph and (b) COF-Na.

Figure R4. FT-IR spectra for (a) COF-Ph and (b) COF-Na.

Figure R5. ^{13}C CP/MAS NMR spectra for (a) COF-Ph and (b) COF-Na.

Figure R6. N₂ Sorption isotherms (77 K), experimental and simulated pore size distribution profiles for COF-Ph (a, b, c) and COF-Na (d, e, f).

(2) In the part of the solid-state ¹³C NMR analysis, we have deleted the statement in the original manuscript according to your comment: "*This result suggests that all the n-butoxyl chains are situated in same pore environment, indicating that they distribute in only one type of pores.*" As for the 2D ¹³C-¹³C NMR spectroscopy you suggested, it is indeed a quite good characterization technique for detection of the mixed linkers in that work (*Nature* **2022**, 606, 706–712). However, this technique could not be applicable for COF-Bu due to the following reason: (i) In the MOF reported in the Nature paper, the two different linkers exhibit distinguishable peaks and thus their correlations could be identified. However, in COF-Bu, the peaks of different butyl chains are indistinguishable. Therefore, it could not tell if correlation signals arise from the carbon atoms in the same butyl chain or from different butyl chains. (ii) The aromatic carbon atoms locating in the triangular pore and the hexagonal pore are also indistinguishable. As a result, the correlation signals (if there is any) between the butyl chain and the benzene ring could not provide useful information for the relative position between them.

Additional results and discussion based on the supplemental experiments have been added to the revised manuscript.

Comment 2: The study on the Janus membrane part seems not complete to me. The authors'

explanation on the wettability of the film's two sides are too arbitrary. I don't see how the pore's hydrophilicity/hydrophobicity transferred to the overall film's property. According to their proposed mechanism, COF-TAB-Hex should display similar properties as COF-Bu (the OH and OR groups can still aligned even though being in the same pore). Besides, many other factors influence the surface wettability, such as the surface morphology. Of all the three COF films this paper present, only COF-Bu shows distinct surface morphology on two sides, which also coincides being the only one with Janus property . The roughness of the bottom surface might contribute more into the hydrophobicity than the surface OR chains do (Colloid and Interface Science Communications 46 (2022) 100556). More experiment evidence is necessary to clarify the Janus properties. It is also interesting to see some applications with COF-Bu membrane to show how the hetero-pore functions exert on the performance.

Response: Thanks for your comment. According to your suggestion, we have conducted additional experiments. Please see the details below.

Firstly, we agree that the hydroxyl and alkyl groups in the COF-TAB-Hex membrane could also experience the induction effect which is similar to that in the COF-Bu membrane. The remarkable difference between them is that the induction effect could be synergistically enhanced by the precise separated alignment of the hydroxyl and alkyl groups in heteropore COF-Bu, but it is unable to achieve in the COF-TAB-Hex membrane. That is because, in COF-TAB-Hex, the hydroxyl groups and alkoxy chains are randomly distributed. Therefore, it is unavoidable that the hydroxyl and alkyl groups of the adjacent layers are aligned in the same side, which might weaken the induction effect due to the opposite orientations of these two substituents. This microscopic induction effect discrepancy might be transferred to the macroscopic morphology difference between COF-Bu and COF-TAB-Hex. The schematic diagram has been shown in Figure 5 in the manuscript.

Further, to check the generality of our hypothesis, we have fabricated COF-Ph membrane through interfacial polymerization (Figure R7a). SEM images reveal that the upper surface of COF-Ph membrane displays a surface with snowflake-like bumps and scattered small sphere-like particles on a relatively smooth surface, while the bottom surface displays a surface with many irregular aggregates. They also display different surface morphology. Through the wettability test, it could be observed that the water contact angles of the upper and bottom sides are 47° and 99° , respectively (Figure R8), indicating its Janus membrane property.

We have also fabricated a POP-Bu membrane which has the same chemical component to COF-Bu but without crystallinity (Figure R7b). Due to its amorphous structure, control over the distribution of side chains is no longer possible. As a result, amphiphilic porosity and synclastic induction effect is inapplicable in the POP-Bu membrane. Although the SEM images revealed some difference for the two sides of the POP-Bu membrane, the water contact angle test indicated that the POP-Bu membrane lost Janus property (Figure R9), in sharp contrast to the COF-Bu membrane. These experimental results could support the conclusion that the amphiphilic porosity of the COFs is a key factor to generate the Janus membranes in this work.

Figure R7. PXRD patterns of the (a) COF-Ph and (b) POP-Bu membranes. The insets show photographs of the corresponding membranes.

Figure R8. SEM images: (a-b) upper surface and (c-d) bottom surface of the COF-Ph membrane; Contact angles: (e) upper surface and (f) bottom surface of the COF-Ph membrane.

Figure R9. SEM images: (a-b) upper surface and (c-d) bottom surface of the POP-Bu membrane; Contact angles: (e) upper surface and (f) bottom surface of the POP-Bu membrane.

As for the application of the COF-Bu membrane, currently we are not able to demonstrate a specific example. In this communication, we are mainly focused on realizing the concept of creating amphiphilic porosity in COFs. In the next stage, we will further explore functions of this novel kind of heteropore COFs.

Additional results and discussion based on the supplemental experiments have been added to the revised manuscript.

Comment 3: In Supporting Information, Figure 2, the NMR spectra looks missing one piece indicated in the red box (it would be better for the authors to label the figures and tables in SI as Figure Sx and Tale Sx to avoid confusions).

Response: Thanks for your kind suggestion. We have completed the NMR spectra and relabeled all the figures and tables as "Figure Sx" and "Table Sx" in revised Supplementary Information.

Comment 4: The pxrd patters of SIOC-COF powder and membrane look different. There seems be a shoulder on the main diffraction peak for the membrane sample (SI, Figure 20). The author should comment on this.

Response: Thank you for pointing out this issue. According to your suggestion, we have pointed out the shoulder peak in revised Supplementary Information and provided an explanation: (page 26, Figure S34) *"Note: The small shoulder on the main diffraction peak of the SIOC-COF marked with * could*

probably be attributed to a partial staircase-like stacking of the COF layers synthesized by interfacial-polymerization.^{3,4}". Such a phenomenon was also observed in literature. For the references, please see doi.org/10.1021/jacs.8b08088; doi.org/10.1038/s41467-022-30647-3.

Comment 5: A zoom in C=N vibration area in the FT-IR spectra can better convey the structure information.

Response: Thanks for your suggestion. We have updated the FT-IR spectra in the revised Supplementary Information.

Reviewer #2:

This is an exciting manuscript on "Creating Amphiphilic Porosity in a Two-Dimensional Covalent Organic Framework via Steric-Hindrance-Mediated Precision Hydrophilic-Hydrophobic Microphase Separation." This manuscript has been written well, and the logic behind the work is sound. Therefore, I recommend acceptance per minor revision.

Response: We greatly appreciate the reviewer for the highly positive comment and your recommendation for acceptance after minor revision.

Comment 1: The structures depicted in figure 1 do not showcase the possibility of the disorder. The long-chain aliphatic hydrocarbons will not organize symmetrically. Therefore, figure 1 should be modified, and the considerations of disorder should be presented in the figure.

Response: Thank you for the suggestion. We have modified figure 1 in revised manuscript.

Comment 2: Author wrote "r COFs and membranes for comparative. " I think it should be "comparison".

Response: Thank you for pointing out this issue. We have modified this expression throughout the text according to the comment in revised manuscript.

Comment 3: I wondered if he has compared the properties with some COFs like TpAZo or other Tp-based COFs. How would these membranes feature?

Response: Thank you for your valuable suggestion. Tp-based membranes typically display hydrophilic characteristics owing to their abundant hydrogen bonding sites (10.1021/jacs.7b06640; 10.1016/j.memsci.2022.120799). It is worth to note that the properties of Tp-COF membranes differ significantly from those discussed in our manuscript. First, due to the symmetry of Tp monomer, Tp-based COFs usually are homopore COFs. Secondly, the unique enol-keto interconversion of Tp-based COFs also differentiate them from common imine-linked COFs or other COFs, which further results in the distinct properties. In this manuscript we only focus on imine-linked COFs.

Comment 4: Very little information regarding the SEM and TEM images of the COF or the membranes has been provided. I would request authors to look into that.

Response: Thank you for the comment. According to your suggestion, we have added more discussion on the SEM and TEM images in the revised manuscript.

Reviewer #3:

The communication of Zhao et al. combines a few concepts in the synthesis of COFs: the use of substituents in the linkers to block certain geometries, the use of hydrophobic and hydrophilic building blocks to create materials with both properties (in this case a Janus membrane).

Although these concepts have been discussed in recent literature (both for MOFs and COFs), the creation of a Janus membrane out of one COF structure is rather new.

Response: We are grateful to the reviewer for taking the time to evaluate our work and greatly appreciate the comments.

Comment 1: The authors over-emphasize the fact that the COFs can have a dual pore size distribution: many papers of such systems (typically kgm but also many others) have appeared since then.

Response: Thanks very much for your comment. This work is focused on fabricating COFs with different pore environment, for which heteropore COFs are believed to be ideal scaffolds. The novelty of this work lies in the precise modulation of the pore environment. It is achieved by precisely introducing hydrophilic and hydrophobic segments into different pores of a heteropore COF. Without such a structure, this design would be challenging to realize.

Comment 2: The application of these membranes is missing: how stable are they? are they self-standing (if not, one side of the membrane gets lost of course and the material is no longer a janus membrane). So it would be nice if the author could show a demonstration of its applicability.

Response: We are appreciative of the reviewer's suggestion. These membranes can remain stable in common organic solvents such as THF, acetone, and ethanol, which was evidenced by PXRD and FT-IR results. Additionally, these membranes are self-supporting. The white nylon film under the COF membranes shown in the images in Supplementary Information serves only for the transfer of these membranes from petri dishes with different solvents.

As for the application of the membranes, currently we are not able to demonstrate a specific example. In this communication, we are mainly focused on realizing the concept of creating amphiphilic porosity in COFs. In the next stage, we will further explore functions of this novel kind of heteropore-based COF membranes.

Again, we thank all the reviewers for the constructive suggestions, which have made our manuscript significantly improved.

Sincerely,

Xin Zhao

Reviewer #1 (Remarks to the Author):

The authors have addressed all the reviewers' comments. The manuscript can be accepted for publication in its current form.

Reviewer #2 (Remarks to the Author):

This manuscript can be accepted now. Very nice piece of work.